# Metallic Stent Mesh Coated with Silver Nanoparticles Suppresses Stent-Induced Tissue Hyperplasia and Biliary Sludge in the Rabbit Extrahepatic Bile Duct

**DOI:** 10.3390/pharmaceutics12060563

**Published:** 2020-06-17

**Authors:** Wooram Park, Kun Yung Kim, Jeon Min Kang, Dae Sung Ryu, Dong-Hyun Kim, Ho-Young Song, Seong-Hun Kim, Seung Ok Lee, Jung-Hoon Park

**Affiliations:** 1Department of Biomedical-Chemical Engineering, The Catholic University of Korea, 43 Jibong-ro, Bucheon-si, Gyeonggi 14662, Korea; wrpark@catholic.ac.kr; 2Department of Radiology, and Research Institute of Clinical Medicine of Chonbuk National University-Biomedical Research Institute of Chonbuk National University Hospital, 20 Geonji-ro, Deokjin-gu, Jeonju, Chonbuk 54907, Korea; kky2kkw@gmail.com; 3Biomedical Engineering Research Center, Asan Institute for Life Sciences, Asan Medical Center, 88 Olympic-ro 43-gil, Songpa-gu, Seoul 05505, Korea; miny2208@naver.com (J.M.K.); kryuds@gmail.com (D.S.R.); 4Department of Radiology, Feinberg School of Medicine, and Robert H. Lurie Comprehensive Cancer Center, Northwestern University, Chicago, IL 60611, USA; dhkim@northwestern.edu; 5Department of Radiology, UT Health Science Center at San Antonio, 7703 Floyd Curl Drive, San Antonio, TX 78229, USA; songh@uthscsa.edu; 6Division of Gastroenterology, Department of Internal Medicine, and Research Institute of Clinical Medicine of Chonbuk National University-Biomedical Research Institute of Chonbuk National University Hospital, 20 Geonji-ro, Deokjin-gu, Jeonju, Chonbuk 54907, Korea; shkimgi@gmail.com

**Keywords:** silver nanoparticles (AgNPs), self-expandable metallic stent (SEMS), tissue hyperplasia, biliary sludge, anti-bacterial response

## Abstract

Recent therapeutic strategies to suppress restenosis after biliary stent placement are insufficient. Here, we demonstrate the usefulness of a self-expandable metal stent (SEMS), a stent mesh coated with silver nanoparticles (AgNPs), for suppression of both stent-induced tissue hyperplasia and biliary sludge formation in the rabbit bile duct. The AgNP-coated SEMSs were prepared using a simple bio-inspired surface modification process. Then, the prepared SEMSs were successfully placed in 22 of 24 rabbits. Sludge formation in the AgNP-coated SEMS groups was significantly decreased compared to the control group on gross findings. Cholangiographic and histologic examinations demonstrated significantly decreased tissue hyperplasia in the AgNP-coated SEMS groups compared with the control group (*p* < 0.05 for all). There were no differences between the AgNP-coated SEMS groups (*p* > 0.05 for all). However, in the group coated with the greatest concentration of AgNPs (Group D), submucosal fibrosis was thicker than in the other AgNP-coated groups (*p* < 0.05 for all). The AgNP-coated metallic stent mesh significantly suppressed stent-induced tissue hyperplasia and biliary sludge formation in the rabbit bile duct. Taken together, the AgNP coating strategy developed in this study could be widely utilized in non-vascular medical devices for anti-bacterial and anti-inflammatory responses.

## 1. Introduction

Self-expandable metal stent (SEMS) placement is a widely adopted treatment for palliation of a malignant biliary obstruction that improves patients’ quality of life and relieves symptoms of jaundice [1,2,3,4]. However, SEMSs are susceptible to stent re-obstruction by tissue hyperplasia and/or tumor ingrowth or overgrowth because of mechanical stress on the adjacent tissue caused by the uncovered SEMS mesh framework or both ends of the covered SEMS [5]. Tissue growth through the stent mesh is also the main obstacle to successful SEMS removal for exchange purposes [6]. Another critical cause of SEMS re-obstruction is biliary sludge, which is induced by a bacterial biofilm on the stent meshes [6,7]. Various types of stents have been investigated to overcome the current drawbacks of biliary stents, mainly focusing on design and shape modification and drug elution [8,9,10,11,12]. However, current stenting strategies are insufficient in terms of clinical effectiveness, and further investigations are required to improve clinical outcomes of biliary stenting for benign or malignant biliary obstruction.

Silver nanoparticles (AgNPs) possess practical anti-bacterial and anti-inflammatory effects and have been widely used in medicine, especially in orthopedic implantable devices, to prevent infections [13,14,15]. In 1992, an in vitro study [16] reported that a silver coating on a polyurethane stent material was effective in suppressing stent restenosis caused by bacterial adherence and subsequent biofilm formation. Although several studies have reported the anti-bacterial and anti-inflammatory effects of AgNP-coated plastic stents and Ag particle-integrated silicone membranes for covered SEMS, no studies have used AgNPs to coat the stent wire itself [17,18,19]. We hypothesized that AgNP coating of a nitinol wire, which is a common material used for biliary SEMSs, might effectively suppress biliary sludge formation and stent-induced inflammatory reactions after SEMS placement. The purpose of this study was thus to investigate the effects of an AgNP-coated metallic stent mesh Scheme 1 for suppression of stent-induced tissue hyperplasia and biliary sludge formation in the rabbit extrahepatic bile duct.

## 2. Materials and Methods

### 2.1. Materials

Dopamine hydrochloride and silver nitrate (AgNO_3_) were purchased from Sigma–Aldrich (St. Louis, MO, USA). Tris(hydroxymethyl)aminomethane hydrochloride was purchased from TCI-Korea (Seoul, Korea).

### 2.2. Preparation of AgNP-Coated SEMS

Each stent was knitted from a thread of 0.110 mm nitinol wire. The stents were 4 mm in diameter and 10 mm in length and had radiopaque markers at both ends. The AgNP coating was added to the surface of nitinol SEMSs using polydopamine (PDA) chemistry (Appendix A) [20]. Dopamine hydrochloride (1 mg/mL) was dissolved in 15 mL of 5 mM Tris buffer (pH 8.5). Then, the SEMSs were immersed in the dopamine solution at room temperature (RT) for 12 h. The PDA-coated SEMSs were washed with deionized water (DW), and the coating procedure was repeated twice under the same conditions. Next, AgNPs were coated onto the surface of the PDA-coated SEMSs [21]. The PDA-coated SEMSs were immersed in 15 mL of AgNO_3_ solution (3, 6, and 12 mg/mL). The reaction proceeded at RT (15 min). After the reaction, the AgNP-coated SEMSs were washed with DW and dried at RT (24 h). 

### 2.3. Surface Characterization of AgNP-Coated SEMS

The surface characteristics were analyzed by scanning electron microscopy (SEM, JSM-820, JEOL Ltd., Tokyo, Japan) with energy-dispersive X-ray spectroscopy (EDS, INCAx-sight, Oxford Instruments, Abingdon, Oxfordshire, UK). The samples, which were control, PDA-coated, and AgNP-coated SEMSs, were fixed on an aluminum pin stub mount, and the elemental mapping of silver and the morphology of the AgNPs on the surface of SEMS were examined. 

### 2.4. In Vitro Cytotoxicity Test

Cytotoxicity tests of control, PDA-coated, and AgNP-coated nitinol wires were performed using a standard Cell Counting Kit-8 (CCK-8) (Dojindo Laboratories, Kyoto, Japan). L929 and 293 cells at a concentration of 1 × 10^4^ cells were plated in each well of a 96-well plate and cultured at 37 °C for 24 h. The three types of wires were placed in different wells for 1, 12, 24, 36, and 48 h, after which 10 µL of CCK-8 solution was added to each well of the 96-well plate, and the cells were incubated. Cell viability was measured using a colorimetric CCK-8 assay (Dojindo Laboratories, Kyoto, Japan). The cell experiments were performed four independent times and statistically analyzed.

### 2.5. Animal Study

This study was approved by the Institutional Animal Care and Use Committee of our institution and conformed to the US National Institutes of Health guidelines for humane handling of laboratory animals. 

The number of animals used to assess the hypothesized difference in tissue hyperplasia after stent placement between control and AgNP-coated SEMSs had been calculated prospectively. The mean percentage of granulation tissue area of the stented rat bile duct on histological examination at 4 weeks after stent placement was expected to be around 50%, with a standard deviation of 10% based on the results of our pilot study (unpublished data). We hypothesized that this area would be decreased by at least 10% in the percentage of granulation tissue area of the stented bile duct with AgNP-coated SEMS. We calculated that a total of 24 animals (6 per group) would be required to detect this difference with statistical significance between the groups, with an alpha level of 0.05 and a beta level of 0.80.

A total of 24 New Zealand rabbits (weight range, 1.5–1.8 kg; Orient Bio, Seongnam, Korea) were randomly divided into four groups by computer-generated random numbers as follows: Group A (*n* = 6) received control SEMSs. Groups B (*n* = 6), C (*n* = 6), and D (*n* = 6) received SEMSs that were AgNP-coated by immersion in 3, 6, and 12 mg/mL of AgNO_3_, respectively (Appendix A). Body weight was measured weekly until sacrifice of the rabbits. All rabbits were euthanized four weeks after the procedure. 

### 2.6. Stent Placement

Anesthesia was induced by intramuscular injection. A 3 cm midline incision on the epigastrium was performed. The stomach and duodenum were gently retracted upward, and the ampulla of Vater was identified grossly (Appendix A). An appropriate puncture site on the duodenum within 2 cm of the orifice of the ampulla of Vater was selected by avoiding vessels. The duodenum was punctured using a 20-gauge angiocatheter with needle (BD Angiocathe Plus; Becton Dickinson, Korea). After confirming that the duodenal wall was penetrated, the central needle was removed, while the angiocatheter was held in place. A catheter tip was manipulated into the orifice of the ampullar of Vater. Then, a 30 cm micro-guidewire was inserted through the angiocatheter into the common bile duct under fluoroscopic guidance. The angiocatheter was advanced over the guidewire across the ampulla of Vater until its tip reached the mid-portion of the common bile duct. With the angiocatheter in place, the guidewire was removed (Appendix A). A compressed stent was loaded in the angiocatheter and positioned in the common bile duct using a pusher rod (Appendix A). The stent was deployed in the common bile duct by retracting the angiocatheter while the pusher rod was held in place. Cholangiography was performed immediately after stent placement to check stent patency and confirm possible complications (Appendix A). The angiocatheter and pusher rod were pulled out from the common bile duct after stent placement. The incision site was sequentially closed with sutures. Antibiotics and analgesics were routinely used for three days after the surgical procedure.

### 2.7. Cholangiographic Examination

Follow-up cholangiography was performed just before sacrifice in all rabbits to check stent position and patency. After midline incision, peritoneal adhesion was gently dissected, and an angiocatheter was placed in the orifice of the ampulla of Vater in the same way. The luminal diameter of the stented extrahepatic bile duct was measured on each cholangiography using Photoshop software (version 6.0; Adobe Systems, Palo Alto, CA, USA). The analyses of the cholangiographic findings were based on the consensus of three observers blinded to group assignment. 

### 2.8. Gross and Histological Examination

After cholangiography, surgical exploration of the liver, biliary duct, and duodenum was done. The degree of biliary sludge and granulation tissue formation was evaluated by gross examination. Stented tissues were fixed in formalin for 24 h. The stented extrahepatic bile duct was transversely sectioned at the proximal and distal regions. The cut stent wires with biliary sludge were carefully removed from the specimens. The degree of biliary sludge formation was evaluated. The slides were stained with Hematoxylin-Eosin (H&E) and Masson’s Trichrome (MT). Histological evaluation using H&E included determination of the degree of submucosal inflammatory cell infiltration, the thickness of submucosal fibrosis, and the granulation tissue-related percentage of the bile duct cross-sectional area of stenosis = 100 × (1 − (stenotic stented area/original stented area)). The degree of inflammatory cell infiltration was subjectively determined according to the distribution and density of the inflammatory cells (1, mild; 2, mild to moderate; 3, moderate; 4, moderate to severe; and 5, severe). The average values of the thickness of submucosal fibrosis and the degree of inflammatory cell infiltration represented the average value of eight points around the circumference [5,20]. The degree of collagen deposition was determined using MT-stained sections. The extent of collagen deposition was subjectively determined, where 1 = mild, 2 = mild to moderate, 3 = moderate, 4 = moderate to severe, and 5 = severe. Histological analysis of the bile duct was performed using a BX51 microscope (Olympus, Tokyo, Japan). Image-Pro Plus software (Media Cybernetics, Silver Spring, MD, USA) was used for the measurements. The analyses of the histological findings were based on the consensus of three observers blinded to group assignment. 

### 2.9. Statistical Analysis

The Kruskal–Wallis or Mann–Whitney U test was used to analyze the differences between groups, as appropriate. *p* < 0.05 was considered statistically significant. If a *p*-value was lower than 0.05, a Bonferroni-corrected Mann–Whitney U test was used to detect the group causing the differences (*p* < 0.008 as statistically significant). Statistical analyses were performed using SPSS software (version 24.0; SPSS, IBM, Chicago, IL, USA).

## 3. Results

### 3.1. Characterization of AgNP-Coated SEMS

AgNP-coated SEMSs were prepared using a bio-inspired surface modification process (Scheme 1) [20]. The PDA layer was first coated onto the SEMS surface, and then AgNPs were grown on the PDA surface. Scanning electron microscope (SEM) analysis was performed to analyze the coating layer modified with AgNPs (Figure 1). The stent coated with PDA showed a smooth shape, similar to the control group, but it became rougher after coating with AgNPs. The surface analysis done with EDS showed that the carbon content was increased compared to that of the bare nitinol surface after coating with PDA, and that the Ag content was significantly increased after coating with AgNPs (Figure 1a–c). In a high-magnification electron microscope image, spherical and plate-shaped AgNPs were observed on the stent surface (Figure 1d,e). The size of the AgNPs varied from 20 to 500 nm. Interestingly, the amount of AgNPs coated on the surface was also increased depending on the amount of AgNO3 used (Appendix A).

The cytotoxicity of the control nitinol wire, PDA-coated wire, and AgNP-coated wire as assessed with L929 and 293 cells showed that no cell death was observed, which indicated that the control, PDA-coated, and GNP-SEMSs were nontoxic in vitro, as shown in Appendix A.

### 3.2. Procedural Outcomes

SEMSs were successfully placed in 22 (91.7%) of the 24 rabbits. Extrahepatic bile duct perforation occurred in two rabbits during the insertion of the stent delivery system (one in group B and one in group D). The remaining 22 rabbits survived until the end of the study without stent-related adverse events. There were no significant differences in body weights between the groups at four weeks (2.43 ± 0.34 kg in group A vs. 2.39 ± 0.29 kg in group B vs. 2.46 ± 0.32 kg in group C vs. 2.41 ± 0.35 kg in group D; *p* = 0.632, Kruskal–Wallis test).

### 3.3. Cholangiographic Findings

Follow-up cholangiography was successfully performed in 20 (90.9%) of 22 rabbits at four weeks after stent placement. Gross detection of the ampulla of Vater failed in two rabbits (one in group A and one in group D) because of severe peritoneal adhesion after stent placement. The mean luminal diameters of the stented extrahepatic bile ducts were significantly different between the groups (*p* < 0.001, Kruskal–Wallis test). The mean (± standard deviation) luminal diameter in group A (2.34 ± 0.23 mm) was much smaller than those in groups B (3.39 ± 0.26 mm), C (3.41 ± 0.21 mm), and D (3.27 ± 0.19 mm) (all *p* < 0.001). The mean luminal diameters were not significantly different between groups B, C, and D (B vs. C, *p* = 0.861; B vs. D, *p* = 0.138; and C vs. D, *p* = 0.061) (Figure 2).

### 3.4. Gross and Histological Findings

The gross and histological findings are shown in Figure 3 and Figure 4. Biliary sludge formation was more prominent in group A than in groups B, C, and D (Figure 3). However, biofilm formation adjacent to the SEMS was observed in all groups. 

The mean percentage of granulation tissue area (Figure 4a,b), the mean thickness of submucosal fibrosis, the degree of inflammatory cell infiltration, and the degree of collagen deposition were significantly different between the groups (all variables *p* < 0.001, Kruskal–Wallis test). The mean percentage of granulation tissue area was significantly higher in group A (56.92 ± 9.80%) than in groups B (21.64 ± 5.61%), C (22.69 ± 6.85%), and D (26.01 ± 7.06%) (all *p* < 0.001). However, there were no statistically significant differences between AgNP-coated SEMS groups in the percentage of granulation tissue area (B vs. C, *p* = 0.712; B vs. D, *p* = 0.143; and C vs. D, *p* = 0.301). The mean thickness of submucosal fibrosis was also significantly higher in group A than in groups B, C, and D (0.79 ± 0.20 mm vs. 0.29 ± 0.10 mm, 0.31 ± 0.09 mm, and 0.39 ± 0.13 mm, respectively; *p* < 0.001 for all). The thickness of submucosal fibrosis was significantly greater in group D than in groups B and C (*p* = 0.012 and *p* = 0.036, respectively). The grade of inflammatory cell infiltration (3.35 ± 0.75 vs. 2.45 ± 0.68, 2.70 ± 0.66, and 2.85 ± 0.74; *p* < 0.001, *p* = 0.006, and *p* = 0.040, respectively) and the degree of collagen deposition (3.45 ± 0.69 vs. 2.75 ± 0.72, 2.85 ± 0.67, and 2.75 ± 0.64; *p* = 0.003, *p* = 0.008, and *p* = 0.002, respectively) were significantly higher in group A than in groups B, C, and D. There were no differences between AgNP-coated SEMS groups in inflammatory cell infiltration (B vs. C, *p* = 0.247; B vs. D, *p* = 0.085; and C vs. D, *p* = 0.504) or in the degree of collagen deposition (B vs. C, *p* = 0.651; B vs. D, *p* > 0.999; and C vs. D, *p* = 0.632).

## 4. Discussion

Management of benign or malignant biliary strictures using covered or uncovered SEMS is currently limited by the development of tissue hyperplasia through the SEMS mesh response to mechanical injury and biofilm formation with biliary sludge adjacent to the placed SEMS. In this study, we were able to coat AgNPs onto the surface of nitinol wire. The PDA coating can reduce silver nitrate to coat AgNPs on its surface [22,23,24]. The distribution of AgNPs coated on the PDA surface was similar to previous studies [25,26,27]. The shape of the coated-AgNPs varied from sphere to plate. The particle size of AgNPs also showed non-uniform size ranging from 20 to 500 nm. The shape and size of AgNPs on the surface can be influenced by synthetic conditions such as PDA coating thickness, buffer pH, temperature, and silver nitrate concentration. Biological effects on the shape and size of AgNPs need to be further investigated in future studies. Various nano- and micro-particles have been extensively researched for antimicrobial, anti-inflammatory, antioxidant, and anticancer effect [28,29,30]. Among them, the AgNPs are well known to have anti-bacterial and anti-inflammatory activities [22,23,24,28,29,30]. In our in vivo study, the degree of stent-induced tissue hyperplasia was significantly less in the AgNP-coated SEMS groups than in the control group at four weeks after stent placement, which correlated with the histological findings. Biliary sludge in the AgNP-coated SEMS groups also prominently decreased compared with that seen in the control group. In aggregate, the results of our study demonstrated that AgNP-coated SEMS successfully suppressed stent-induced tissue hyperplasia and biliary sludge formation in the rabbit extrahepatic bile duct.

AgNPs, which have a diameter in the range of 1–100 nm, are the most frequently used nanomaterial because of their broad spectrum of useful properties, which include antimicrobial effects against various pathogens, such as bacteria, fungi, and viruses, as well as anti-inflammatory activity [31,32]. However, various properties of AgNPs, such as size, distribution, agglomeration, and dissolution rate, which meditate the advantages of this nanotechnology, also influence their potential cytotoxicity and pro-inflammatory effects [33,34,35]. Although AgNPs have emerged as promising nanomaterials for biomedical applications, they have potential toxicity risks [36]. The results of the cell viability test with the AgNP-coated wire did not show significant toxicity. According to previous studies [37,38], AgNPs with a size smaller than 50 nm showed more significant toxicity than nanoparticles with a larger size. Additionally, the cytotoxicity of AgNPs was dependent on the nanoparticle concentration regardless of the particle size. The results were reported to be due to oxidative stress caused by AgNPs. However, further studies are required to investigate the long-term toxicity of AgNPs and the possibility of AgNPs release from the surface of the functionalized SEMS. In our study, three different Ag concentrations (3, 6, and 12 mg/mL) were used in the coating process to evaluate the dose-range effect of AgNPs. The results of our in vivo study demonstrate that the thickness of submucosal fibrosis in the relatively high Ag concentration group (12 mg/mL) was higher than that of the 3 and 6 mg/mL Ag concentration groups. Furthermore, the degree of inflammatory cell infiltration in the AgNP-coated SEMS groups gradually increased in an AgNPs dose-dependent manner. Exposure to a higher concentration of AgNPs might induce a proliferative response, i.e., increased thickness of submucosal fibrosis caused by the pro-inflammatory effects of AgNPs. Although further cytotoxicity studies are needed to investigate AgNP dose-range effects, our findings suggest that moderate Ag concentrations (3 and 6 mg/mL) have favorable effects on inhibition of granulation tissue formation after stent placement and minimization of the pro-inflammatory effects of AgNPs in the rabbit extrahepatic bile duct.

AgNPs have been applied to stents using various methods, such as Ag and polyurethane (PU) composite suspensions [17], Ag-loaded heparin/chitosan multilayer films [18], and Ag-particle integrated silicone membranes [19]. The various types of plastic or covered stents with Ag significantly reduced bacterial adherence and prolonged stent patency in vitro and in vivo in experimental studies [16,17,18,19]. In our study, AgNP-coated SEMSs were prepared simply, using PDA as the Ag anchor. The surface of the bare SEMS was first modified using PDA, which helps AgNPs adhere to the surface of the nitinol wire, and then AgNPs were attached to the PDA-coated surface of the stent. PDA has been reported to be a promising adherent coating on a wide range of organic or inorganic substrates because of its ability for mussel-inspired self-polymerization [39,40]. The bioinspired PDA coating has been used as a binding layer to improve the biocompatibility and adherence of biomaterial surfaces [39,40,41]. This synthesis method is straightforward and can be completed in a short time, providing the opportunity for large-scale synthesis of AgNP-coated SEMSs. The excellent productivity and stability of the AgNP-coated SEMS provide opportunities for clinical applications. Placement of AgNP-coated SEMSs in patients with malignant and benign biliary strictures will not only open the stricture site of biliary trees but also suppress tissue hyperplasia and biliary sludge after stent placement better than bare metallic stents can.

In the present study, a newly developed rabbit biliary stent model was used. Biliary stents were successfully placed in the extrahepatic bile duct of rabbits by simple laparotomy and duodenostomy using a 20-gauge angiocatheter. Until now, to the best of our knowledge, a rabbit biliary model of stent-induced tissue hyperplasia and biliary sludge formation with the use of SEMS has not been fully described. Our model has several advantages as an in vivo study. First, this model has advantages in imaging analysis. Initial cholangiography was easily performed before and immediately after stent placement, and follow-up cholangiography was successful in most rabbits because the degree of peritoneal adhesion was not severe. Second, a rabbit model is more conducive to manipulation than are large animal models, such as canine or porcine models. The use of rabbits also provides economic benefits, and the size of the study population can be increased quickly in order to increase the power of statistical analysis. Third, stent migration did not occur during the follow-up period, whereas stent-induced tissue hyperplasia and biliary sludge were well developed within four weeks after stent placement. Our model would be useful for studying the mechanism of biliary sludge formation and the pathologic processes of stent-induced tissue hyperplasia, and to evaluate therapeutic strategies for these phenomena in the future.

Advances in stent technologies have resulted in fewer complications and high technical success rates; however, prolonged stent patency remains technically challenging. SEMSs provided a relatively longer duration of stent patency compared with plastic stents [42]. Uncovered SEMSs occlude the stent lumen by tissue ingrowth through the wire mesh. For this reason, covered SEMSs were introduced to prolong stent patency by preventing tissue ingrowth [3,43,44]. However, covered SEMSs carry increased risks of migration as well as cholecystitis and/or pancreatitis by blocking the cystic or pancreatic ducts [3,42,43,44,45,46]. Our therapeutic strategy using AgNP-coated SEMSs can be applied to uncovered SEMSs to prevent tissue ingrowth and may prolong stent patency by decreasing tissue hyperplasia and biliary sludge formation.

Our study has a number of limitations. First, the AgNP-coated SEMSs were placed in animals with normal extrahepatic bile ducts; wound healing after mechanical injury secondary to stent placement may differ considerably. Second, we did not assess the cytotoxicity of AgNPs and could not carry out a quantitative analysis of biliary sludge formation after stent placement. Nevertheless, although additional studies are required to evaluate the pre-clinical effects and safety of these stents, the results of our study support the basic concept of using AgNP-coated SEMSs to suppress stent-induced tissue hyperplasia and biliary sludge after stent placement in the rabbit extrahepatic bile duct. 

## 5. Conclusions

In this study, SEMSs with AgNP-coated stent mesh were successfully fabricated through PDA chemistry to suppress restenosis of the biliary stent. SEMSs with AgNP-coated stent mesh significantly suppressed stent-induced tissue hyperplasia and biliary sludge formation in the rabbit common bile duct. The AgNP coating strategy used in this study is expected to be applicable to the anti-bacterial and anti-inflammatory properties of various medical devices.

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
