# Peer review of "Metallic Stent Mesh Coated with Silver Nanoparticles Suppresses Stent-Induced Tissue Hyperplasia and Biliary Sludge in the Rabbit Extrahepatic Bile Duct"

_pharmaceutics, 2020, doi:10.3390/pharmaceutics12060563_

Round 1

Reviewer 1 Report

Presented article is written in a good style, logicaly arranged. There it is possible to notice small mistakes in English language or missprints. The results are summarized widely enough and chosen methods are appropriate. My reccomandation to the improvement is to disscus widely the effect of the size and shape of AgNPs. The second notice - confirm (based on citations or previous work) that the number of 24 rabbits is sufficient for truly support od obtained results and their meaning.

Author Response

Reviewer# 1:

Q1) Presented article is written in a good style, logically arranged. There it is possible to notice small mistakes in English language or misprints. The results are summarized widely enough and chosen methods are appropriate. My recommendation to the improvement is to discuss widely the effect of the size and shape of AgNPs.

Response: We appreciate your comment. The PDA coating can reduce silver nitrate to attach AgNPs on its surface. The shape of the coated-AgNPs varied from sphere to plate. The shape of AgNPs can be influenced by synthetic conditions such as PDA coating thickness, buffer pH, temperature, and silver nitrate concentration. Biological effects on the shape of AgNPs need to be further investigated in future studies.

(In line 16-19 on page 10) The PDA coating can reduce silver nitrate to attach AgNPs on its surface.22-24 The shape of the coated-AgNPs varied from sphere to plate. The shape of AgNPs can be influenced by synthetic conditions such as PDA coating thickness, buffer pH, temperature, and silver nitrate concentration.

(In the references)

(22) Luo, H.; Gu, C.; Zheng, W.; Dai, F.; Wang, X.; Zheng, Z. Facile synthesis of novel size-controlled antibacterial hybrid spheres using silver nanoparticles loaded with poly-dopamine spheres. RSC Advances 2015, 5 (18), 13470-13477.

(23) Liu, L.; Cai, R.; Wang, Y.; Tao, G.; Ai, L.; Wang, P.; Yang, M.; Zuo, H.; Zhao, P.; He, H. Polydopamine-assisted silver nanoparticle self-assembly on sericin/agar film for potential wound dressing application. International journal of molecular sciences 2018, 19 (10), 2875.

(24) Guan, M.; Chen, Y.; Wei, Y.; Song, H.; Gao, C.; Cheng, H.; Li, Y.; Huo, K.; Fu, J.; Xiong, W. Long-lasting bactericidal activity through selective physical puncture and controlled ions release of polydopamine and silver nanoparticles–loaded TiO2 nanorods in vitro and in vivo. International journal of nanomedicine 2019, 14, 2903.

Q2) The second notice - confirm (based on citations or previous work) that the number of 24 rabbits is sufficient for truly support od obtained results and their meaning.

Response: Thank you for your comment. We have revised the Materials and Methods section regarding how to calculate sample size for our animal study as follows.

(In line 13-20 on page 5) The number of animals used to assess the hypothesized difference in tissue hyperplasia after stent placement between control and AgNP-coated SEMSs had been calculated prospectively. The mean percentage of granulation tissue area of the stented rat bile duct on histological examination at 4 weeks after stent placement was expected to be around 50 % with a standard deviation of 10% based on results of our pilot study (unpublished data). We hypothesized that this area would be decreased by at least 10% in the percentage of granulation tissue area of the stented bile duct with AgNP-coated SEMS. We calculated that a total of 24 animals (6 per group) would be required to detect this difference with statistical significance between the groups, with an alpha level of .05 and a beta level of .80.

Thank you very much for the appropriate and valuable review comments. We are sure that these comments improved the quality of the manuscript significantly.

Reviewer 2 Report

The manuscript is quite interesting dealing a very  important application  of silver nanoparticles to SEMS . The use of silver nanoparticles in medicine, especially in implantable devices is an increasing field of research. The manuscript is generally well written but it lacks of a major part, the nanoparticles preparation  and characterization. In the experimental part this section is completely absent and in the results session there is a very short report on sem analysis. The size, homogeneity and in general characterization of Ag nanoparticles is a key point both in the activity and cytoxicity of these nanoparticles. More details also concerning cytotoxic effects of the developed nanoparticles should be reported

Author Response

Reviewer# 2:

Comments and Suggestions for Authors

Q1) The manuscript is quite interesting dealing a very important application of silver nanoparticles to SEMS. The use of silver nanoparticles in medicine, especially in implantable devices is an increasing field of research. The manuscript is generally well written but it lacks of a major part, the nanoparticles preparation and characterization. In the experimental part this section is completely absent and in the results session there is a very short report on sem analysis. The size, homogeneity and in general characterization of Ag nanoparticles is a key point both in the activity and cytotoxicity of these nanoparticles.

Response: Thank you for your comment. We have revised the Materials & Methods section regarding SEM analysis as follows.

(In line 19-24 on page 4) 2.3. Surface characterization of AgNP-coated SEMS, The surface characteristics were analyzed by scanning electron microscopy (SEM, JSM-820, JEOL Ltd., Tokyo, Japan) with energy-dispersive X-ray spectroscopy (EDS, INCAx-sight, Oxford Instruments, Abingdon, Oxfordshire, UK). The sample was fixed on an aluminum pin stub mount and the elemental mapping of silver and the morphology of the AgNPs on the surface of SEMS were examined.

Q2) More details also concerning cytotoxic effects of the developed nanoparticles should be reported

Response: Thank you for your comment. Although AgNPs have emerged as an promising nanomaterials for biomedical applications, that have potential toxicity risks. In the results of cell viability test with the AgNP-coated wire did not show significant toxicity. However, further studies are required to investigate the long-term toxicity of AgNPs and the possibility of AgNPs release from the surface of the functionalized SEMS.

(In line 8-11 on page 11) Although AgNPs have emerged as promising nanomaterials for biomedical applications, that have potential toxicity risks.33 The results of the cell viability test with the AgNP-coated wire did not show significant toxicity. However, further studies are required to investigate the long-term toxicity of AgNPs and the possibility of AgNPs release from the surface of the functionalized SEMS.

(In the references) (33) Ahamed, M.; AlSalhi, M. S.; Siddiqui, M. Silver nanoparticle applications and human health. Clinica chimica acta 2010, 411 (23-24), 1841-1848.

Thank you very much for the appropriate and valuable review comments. We are sure that these comments improved the quality of the manuscript significantly.

Reviewer 3 Report

The manuscript "Metallic Stent Mesh Coated with Silver Nanoparticles Suppresses Stent-induced Tissue Hyperplasia and Biliar Sludge in the Rabbit Extrahepatic bile Duct" describes the use of medical device combined with silver nanoparticles for suppressing tissue hyperplasia in animal models. Rabbit were used as suitable models for this study. The manuscript is well organised and shows some novelty in biomedical devices.

It is opinion of this reviewer that the manuscript can be accepted on Pharmaceutics after minor revision.

Specific comments:

- page 9, lines 240 - 244 "Management of benign or malignant biliary strictures using covered or uncovered SEMS is currently limited by the development of tissue hyperplasia through the SEMS mesh response to mechanical injury and biofilm formation with biliary sludge adjacent to the placed SEMS. In this study, we were able to coat AgNPs onto the surface of nitinol wire. AgNPs are well known to have antibacterial and anti-inflammatory activities.": There are nano and micro- particles making form different materials which have some anti-inflammatory as well as other properties (Cilurzo et al., Curr Drug Targets. 2015;16(14):1612-22; Di Francesco et al., Planta Med. 2017;83(5):482-491; Critello et al., Phlebology. 2019;34(3):191-20). Please include a briefly description of these nano and micro- carriers and include these reference in the paper.

Author Response

Reviewer# 3:

Comments and Suggestions for Authors

The manuscript “Metallic Stent Mesh Coated with Silver Nanoparticles Suppresses Stent-induced Tissue Hyperplasia and Biliary Sludge in the Rabbit Extrahepatic bile Duct” describes the use of medical device combined with silver nanoparticles for suppressing tissue hyperplasia in animal models. Rabbit were used as suitable models for this study. The manuscript is well organized and shows some novelty in biomedical devices.

It is opinion of this reviewer that the manuscript can be accepted on Pharmaceutics after minor revision.

Response: Thank you for your encouragement.

Specific comments:

Q1) page 9, lines 240-244 “Management of benign or malignant biliary strictures using covered or uncovered SEMS is currently limited by the development of tissue hyperplasia through the SEMS mesh response to mechanical injury and biofilm formation with biliary sludge adjacent the placed SEMS. In this study, we were able to coat AgNPs onto the surface of nitinol wire. AgNPs are well known to have antibacterial and anti-inflammatory activities.”: There are nano and micro-particles making form different materials which have some anti-inflammatory as well as other properties (Cilurzo et al., Curr Drug Targets. 2015;16(14):1612-22; Di Francesco et al., Planta Med. 2017;83(5):482-491; Critello et al., Phlebology. 2019;24(3):191-20). Please include a briefly description of these nano and micro-carriers and include these reference in the paper.

Response: Thank you very much for your comment. We have revised the Discussion section with additional three references as follows.

(In line 19-21 on page 10) Biological effects on the shape of AgNPs need to be further investigated in future studies. Various nano and micro-particles have extensively researched for antimicrobial, anti-inflammatory, antioxidant, and anticancer effect.25-27

(In the references)

(25) Cilurzo, F.; Chiara Cristiano, M.; Di Marzio, L.; Cosco, D.; Carafa, M.; Anna Ventura, C.; Fresta, M.; Paolino, D. Influence of the supramolecular micro-assembly of multiple emulsions on their biopharmaceutical features and in vivo therapeutic response. Current drug targets 2015, 16 (14), 1612-1622.

(26) Di Francesco, M.; Primavera, R.; Fiorito, S.; Cristiano, M. C.; Taddeo, V. A.; Epifano, F.; Di Marzio, L.; Genovese, S.; Celia, C. Acronychiabaueri analogue derivative-loaded ultradeformable vesicles: physicochemical characterization and potential applications. Planta medica 2017, 83 (05), 482-491.

(27) Critello, C. D.; Fiorillo, A. S.; Cristiano, M. C.; de Franciscis, S.; Serra, R. Effects of sulodexide on stability of sclerosing foams. Phlebology 2019, 34 (3), 191-200.

Thank you very much for the appropriate and valuable review comments. We are sure that these comments improved the quality of the manuscript significantly.

Round 2

Reviewer 2 Report

Dear authors,

Some key parts of the manuscript still are not addressed.

Concerning Q1,  the  authors introduced in the experimental part “Surface characterization of AgNP-coated SEMS” but of course these data should be discussed also in the results and discussion. The size, homogeneity and in general characterization of Ag nanoparticles is a key point both in the activity and cytotoxicity of these nanoparticles. This is a very sensitive point and in the present form, the paper is not acceptable.

Concerning Q2 it is a very thoughtful point too, please try to better comment your data in particular in the different tested concentrations, size, etc..  with the available data in the literature.

Author Response

Response to the referees’ comments (Manuscript ID: pharmaceutics-815402)

First of all, we would like to thank the reviewers for a thorough reading of the manuscript and helpful comments and suggestions. We have revised our manuscript based on the comments made by the reviewer, and we believe the manuscript is now significantly improved. The followings are the point-by-point responses to the reviewer’s comments.

Reviewer# 2:

Some key parts of the manuscript still are not addressed.

Concerning Q1, the authors introduced in the experimental part “Surface characterization of AgNP-coated SEMS” but of course these data should be discussed also in the results and discussion. The size, homogeneity and in general characterization of Ag nanoparticles is a key point both in the activity and cytotoxicity of these nanoparticles. This is a very sensitive point and in the present form, the paper is not acceptable.

Response: Thank you for your valuable comment. The distribution of AgNPs coated on the PDA surface was similar to previous studies. The shape of the coated-AgNPs varied from sphere to plate. The particle size of AgNPs also showed non-uniform size ranging from 20 to 500 nm.

(In line 7 on Page 8) The size of the AgNPs varied from 20 to 500 nm.

(In line 19-24 on Page 8) The distribution of AgNPs coated on the PDA surface was similar to previous studies.25-27 The shape of the coated-AgNPs varied from sphere to plate. The particle size of AgNPs also showed non-uniform size ranging from 20 to 500 nm. The shape and size of AgNPs on the surface can be influenced by synthetic conditions such as PDA coating thickness, buffer pH, temperature, and silver nitrate concentration. Biological effects on the shape and size of AgNPs need to be further investigated in future studies.

(In the references)

(25) Lu, Z.; Xiao, J.; Wang, Y.; Meng, M. In situ synthesis of silver nanoparticles uniformly distributed on polydopamine-coated silk fibers for antibacterial application. Journal of colloid and interface science 2015, 452, 8-14.

(26) Wang, F.; Han, R.; Liu, G.; Chen, H.; Ren, T.; Yang, H.; Wen, Y. Construction of polydopamine/silver nanoparticles multilayer film for hydrogen peroxide detection. Journal of Electroanalytical Chemistry 2013, 706, 102-107.

(27) Tang, L.; Livi, K. J.; Chen, K. L. Polysulfone membranes modified with bioinspired polydopamine and silver nanoparticles formed in situ to mitigate biofouling. Environmental Science & Technology Letters 2015, 2 (3), 59-65.

Concerning Q2, it is a very thoughtful point too, please try to better comment your data in particular in the different tested concentrations, size, etc..  with the available data in the literature

Response: We appreciate your valuable comment. According to previous studies, AgNPs with a size smaller than 50 nm showed greater toxicity than nanoparticles with a larger size. Additionally, the cytotoxicity of AgNPs was dependent on the nanoparticle concentration regardless of the particle size. The results were reported to be due to oxidative stress caused by AgNPs.

(Line 13-16 on Page 11) According to previous studies,34-35 AgNPs with a size smaller than 50 nm showed more significant toxicity than nanoparticles with a larger size. Additionally, the cytotoxicity of AgNPs was dependent on the nanoparticle concentration regardless of the particle size. The results were reported to be due to oxidative stress caused by AgNPs.

(In the references)

(34) Kim, T. H.; Kim, M.; Park, H. S.; Shin, U. S.; Gong, M. S.; Kim, H. W. Size‐dependent cellular toxicity of silver nanoparticles. Journal of biomedical materials research Part A 2012, 100 (4), 1033-1043.

(35) Park, M. V.; Neigh, A. M.; Vermeulen, J. P.; de la Fonteyne, L. J.; Verharen, H. W.; Briedé, J. J.; van Loveren, H.; de Jong, W. H. The effect of particle size on the cytotoxicity, inflammation, developmental toxicity and genotoxicity of silver nanoparticles. Biomaterials 2011, 32 (36), 9810-9817.

Thank you very much for the appropriate and valuable review comments. We are sure that these comments improved the quality of the manuscript significantly.